# Circulating tumour DNA reflects treatment response and clonal evolution in chronic lymphocytic leukaemia

Paul Yeh[1,2,3,*], Tane Hunter[1,3,*], Devbarna Sinha[1,*], Sarah Ftouni[1], Elise Wallach[1], Damian Jiang[4], Yih-Chih Chan[1], Stephen Q. Wong[1], Maria Joao Silva[1], Ravikiran Vedururu[1], Kenneth Doig[1,3], Enid Lam[1,3], Gisela Mir Arnau[1], Timothy Semple[1], Meaghan Wall[5,6], Andjelija Zivanovic[1,3], Rishu Agarwal[1,3], Pasquale Petrone[1], Kate Jones[1], David Westerman[1,2,3], Piers Blombery[1,2], John F. Seymour[2,3], Anthony T. Papenfuss[1,3,7], Mark A. Dawson[1,2,3,8], Constantine S. Tam[2,3,6,9] & Sarah-Jane Dawson[1,2,3,8]

Several novel therapeutics are poised to change the natural history of chronic lymphocytic leukaemia (CLL) and the increasing use of these therapies has highlighted limitations of traditional disease monitoring methods. Here we demonstrate that circulating tumour DNA (ctDNA) is readily detectable in patients with CLL. Importantly, ctDNA does not simply mirror the genomic information contained within circulating malignant lymphocytes but instead parallels changes across different disease compartments following treatment with novel therapies. Serial ctDNA analysis allows clonal dynamics to be monitored over time and identifies the emergence of genomic changes associated with Richter's syndrome (RS). In addition to conventional disease monitoring, ctDNA provides a unique opportunity for non-invasive serial analysis of CLL for molecular disease monitoring.

[1] Division of Cancer Research, Peter MacCallum Cancer Centre, 305 Grattan Street, Melbourne, Victoria 3000, Australia. [2] Division of Cancer Medicine, Peter MacCallum Cancer Centre, Melbourne, Victoria 3000, Australia. [3] Sir Peter MacCallum Department of Oncology, University of Melbourne, Melbourne, Victoria 3000, Australia. [4] Department of Radiology, Peter MacCallum Cancer Centre, Melbourne, Victoria 3000, Australia. [5] Victorian Cancer Cytogenetics Service, St Vincent's Hospital, Fitzroy, Victoria 3065, Australia. [6] Department of Medicine, St Vincent's Hospital, University of Melbourne, Fitzroy, Victoria 3065, Australia. [7] Walter and Eliza Hall Institute of Medical Research, Parkville, Victoria 3000, Australia. [8] Centre for Cancer Research, University of Melbourne, Parkville, Victoria 3000, Australia. [9] Department of Haematology, St Vincent's Hospital, Fitzroy, Victoria 3065, Australia. * These authors contributed equally to this work. Correspondence and requests for materials should be addressed to S.J.D. (email: Sarah-Jane.Dawson@petermac.org).

Major therapeutic advances in chronic lymphocytic leukaemia (CLL) have occurred in recent years. The advent of monoclonal antibodies directed against CD20 and CD52 heralded the beginning of the targeted therapy revolution[1] and these antibodies have since been joined by small molecule inhibitors, which target phosphoinositide 3-kinase (idealisib)[2], the Bruton's tyrosine kinase (ibrutinib)[3–5] and BH3 mimetics, which inhibit BCL-2 (Venetoclax and Navitoclax)[6–8]. Although highly effective, the increasing use of these therapies has revealed deficiencies in current strategies for disease monitoring, largely due to differential treatment responses across separate disease compartments. New opportunities for molecular disease monitoring are now being provided from our increased understanding of the genomic landscape of CLL[9,10]. Recent studies have uncovered recurrent somatic mutations in several genes[11–14], many of which are associated with high-risk disease and therapy resistance[12,14,15]. Moreover, therapeutic selection pressure has been shown to lead to marked clonal evolution[10,16] and potentially increase the development of Richter's syndrome (RS)[7,17]. The challenge of monitoring multiple disease subclones across different disease compartments has highlighted the need for improved methods to accurately assess treatment response, allow the early identification of therapeutic failure and follow clonal dynamics over time. For this reason, we sought to assess the role of circulating tumour DNA (ctDNA) as a molecular tool for disease monitoring in CLL. We show that ctDNA dynamics reflect changes in disease across different tissue compartments. Furthermore, ctDNA is able to represent new genomic changes that occur in the context of disease progression.

## Results

**Clinical cohort**. Our analysis was undertaken in 32 patients with relapsed/refractory CLL commencing 1 of 2 novel therapies; ibrutinib or venetoclax (Supplementary Table 1). Targeted amplicon deep sequencing (TS) across a panel of the seven most frequently mutated genes in CLL (*SF3B1*, *NOTCH1*, *ATM*, *TP53*, *MYD88*, *KRAS* and *BIRC3*) was performed on DNA collected pretreatment from (i) the peripheral blood (PB) mononuclear layer (MNL) containing the leukaemic cells, (ii) plasma and (iii) matched normal DNA. Using this bespoke panel, somatic driver mutations were identified in 25/32 (78%) individuals in both the matched MNL and plasma DNA samples (Supplementary Table 1). Next, we followed these mutations in serial plasma samples (n = 111) across the cohort using the same TS approach, as well as digital PCR (dPCR) in selected cases, to provide orthogonal validation of the sequencing findings. Quantification of mutant allele fraction (MAF) by either TS or dPCR showed excellent correlation (Supplementary Fig. 1). ctDNA was detectable in all 25 cases and comprised between 0.1 and 90% of total circulating DNA. As a comparator for the somatic gene mutations, patient-specific immunoglobulin heavy chain (IGH) rearrangements were characterized in a subset of patients and specific dPCR assays were designed to follow these in plasma. Importantly, ctDNA levels as assessed by following somatic gene mutations, correlated closely with ctDNA levels as assessed by monitoring specific immunoglobulin heavy chain rearrangements (Fig. 1a–c and Supplementary Fig. 2).

**ctDNA reflects dynamics in different disease compartments**. In clinical practice, CLL tumour burden and treatment response is typically monitored through serial PB lymphocyte (PBL) counts, multi-colour flow cytometry, radiological imaging and bone marrow (BM) biopsies according to the International Workshop on CLL (iwCLL) criteria[18]. The introduction of novel therapies has seen specific patterns emerge, particularly in

relation to changes in PBL counts post treatment. Venetoclax is highly effective at clearing the circulating malignant cells and the PBL count falls rapidly once treatment is initiated[7]. Notably, the rapid PBL clearance and reduction in tissue disease burden following venetoclax treatment was closely paralleled by a decrease in ctDNA levels (Fig. 1a and Supplementary Fig. 3). Conversely, ibrutinib is known to cause a worsening of peripheral lymphocytosis due to the redistribution of tissue-resident CLL cells from the lymph nodes (LN) into the PB[19]. This negates the clinical utility of PBL counts for disease monitoring, as changes will not necessarily correlate with treatment response. Importantly, the initial worsening of peripheral lymphocytosis following ibrutinib therapy was not associated with a sustained parallel rise in ctDNA. Instead, ctDNA levels declined, reflecting the effects of therapy on reducing disease burden assessed radiologically (Fig. 1b and Supplementary Fig. 4). Consistent with this finding, we found a strong correlation between ctDNA levels and changes in the extent of lymphadenopathy on serial radiological imaging across the series (Fig. 1a–d, Supplementary Figs 2–4 and Supplementary Table 2). In contrast, dynamic fluctuations in ctDNA were not as well correlated with MAF changes in MNL DNA and PBL counts (Supplementary Fig. 1 and Supplementary Table 2), highlighting the finding that ctDNA does not simply reflect the circulating malignant disease.

In the clinical management of CLL, an issue that is not addressed adequately with a single test is the ability to simultaneously assess treatment response across all disease sites, including both the circulating and tissue compartments. Across the series, selected somatic mutations were identified in plasma at 99/111 (89%) time points and in MNL DNA at 83/111 (75%) time points with concordance on 89/111 (80%) occasions (Fig. 1e and Supplementary Table 3). Before treatment, the detection of mutations in ctDNA and MNL DNA showed strong concordance across 22/23 (96%) time points (Supplementary Table 3). However, following treatment there were 18/88 (20%) time points where mutations were identified in ctDNA but not in MNL DNA. The majority of these events were in patients with compartmentalized disease (CLL001, CLL004 and CLL050), where the predominant site of disease was in the LNs (Supplementary Table 2, Fig. 1c and Supplementary Figs 3 and 5). Case CLL001 provides an example of this scenario, where despite evidence of a significant treatment response to obinutuzumab in PB and BM, the patient developed increasing lymphadenopathy both clinically and radiologically. This was paralleled by a rise in ctDNA, highlighting the ability of ctDNA to reflect an increase in radiological disease burden following treatment. Following initiation of ibrutinib, the bulky lymphadenopathy was markedly reduced and this compartmentalized disease response was matched by a significant reduction in ctDNA (Fig. 1c). Together, these findings reveal the ability of ctDNA analysis to simultaneously track tumour kinetics across multiple disease sites and suggest that ctDNA may be further useful in cases where disease is predominantly confined to the tissues such as small lymphocytic lymphoma.

In patients who achieve complete responses by conventional criteria, the demonstration of minimal residual disease (MRD) has been shown to be an important prognostic factor[20,21]. The current gold standard for MRD monitoring involves immunophenotyping utilizing flow cytometry on the BM or PB specimens to detect the residual leukaemic clone; however, the analysis is highly dependent on assessment of a sufficient number of cells and the use of optimal antibody combinations[22,23]. Analysis of MRD by multi-colour flow cytometry on the BM or PB was performed at 30 time points across our series at high sensitivity using consensus methodology (Fig. 1f) and the results showed complete concordance with ctDNA analysis in all cases.

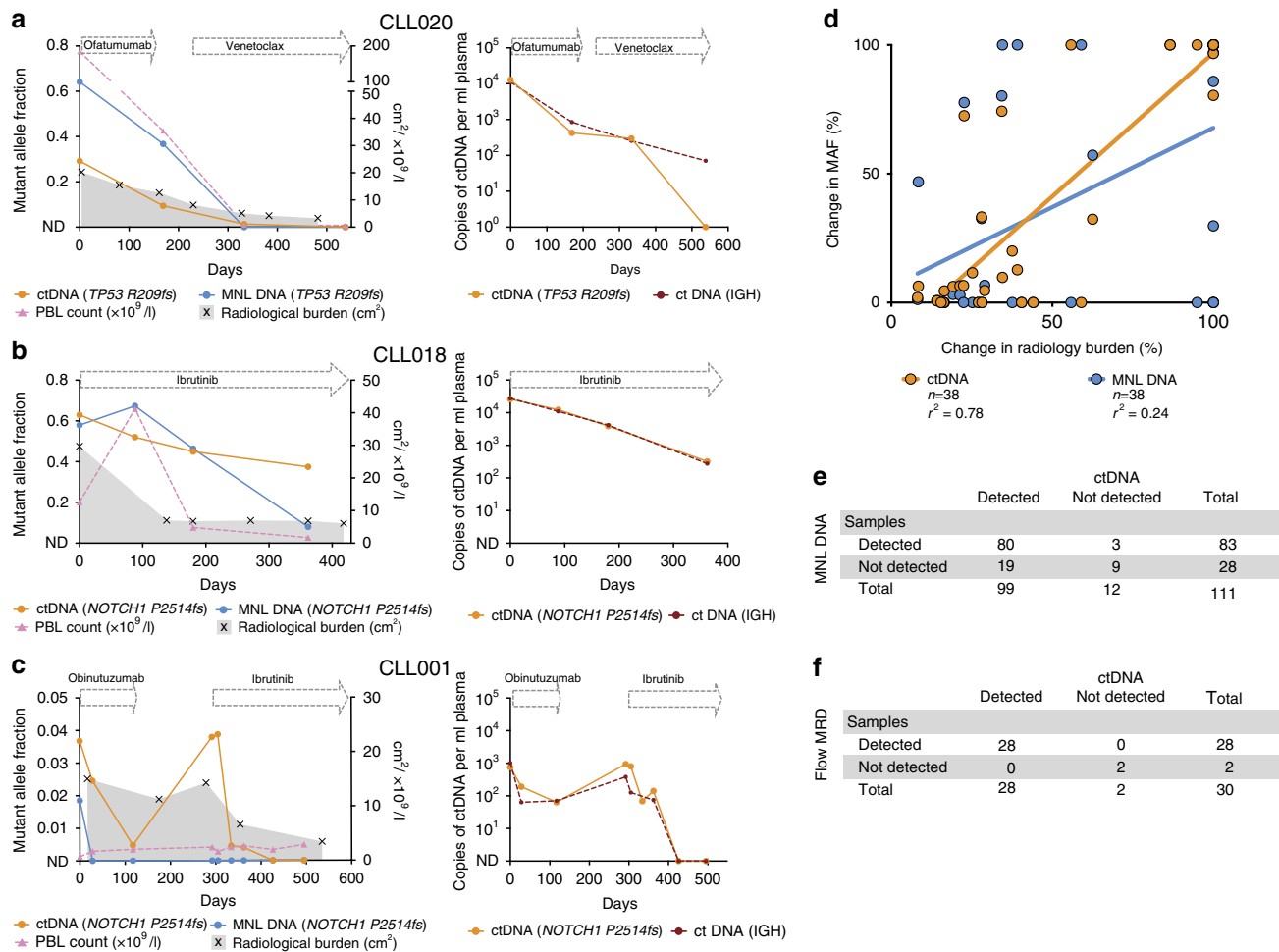

**Figure 1 | ctDNA dynamics reflect changes in disease burden across different tissue compartments.** (**a**) Case CLL020: plasma ctDNA levels by *TP53* p.R209fs mutation (left) and IGH (right) decreased over the course of ofatumumab and venetoclax treatment, which paralleled the decline in mutant DNA levels in the MNL, the decrease in PBL count and the reduction in lymphadenopathy as assessed by imaging. (**b**) Case CLL018: the initial peripheral lymphocytosis observed post ibrutinib treatment coincided with a increase in the abundance of the *NOTCH1* p.P2415fs mutation in the MNL. This contrasted with plasma ctDNA levels of both the *NOTCH1* p.P2415fs mutation (left) and IGH (right), which instead reflected the reduction in radiological disease burden. (**c**) Case CLL001: following obinutuzumab therapy, rapid clearance of the circulating leukaemic cells was observed and the *NOTCH1* p.P2415fs mutation became undetectable in the MNL. This coincided with a parallel decrease in plasma ctDNA as assessed by *NOTCH1* p.P2415fs mutation (left) and IGH (right) levels, although ctDNA did not become undetectable. The patient then had compartmentalized disease progression with increasing lymphadenopathy that was matched by a plasma ctDNA rise. Subsequent ibrutinib treatment resulted in a reduction in lymphadenopathy and plasma ctDNA. These dynamic changes were not reflected in the PBL count or by the *NOTCH1* p.P2415fs mutation in MNL DNA. (**d**) Correlation between the change in ctDNA/MNL MAF versus change in radiological disease burden (cm$^2$) from the maximal value analysed for each individual patient across any time point. Of 38 matched serial time points from a total of 12 patients, the correlation with radiology was significantly better with ctDNA ($r^2 = 0.78$, $P < 0.0001$) than with MNL DNA ($r^2 = 0.24$, $P = 0.0019$). (**e**) Comparison of ctDNA and MNL detection across 111 matched time points. (**f**) Comparison of ctDNA with detection of CLL by multi-colour flow cytometry on PB or BM.

**ctDNA detects clonal evolution in CLL patients with RS.** We next sought to evaluate the role of ctDNA in monitoring disease progression and clonal evolution in CLL under the selective pressure of novel therapies. Recent studies have reported the development of RS as a mechanism of relapse in patients receiving novel therapies for CLL[7,17,24]. RS is usually manifested by disease transformation into diffuse large B-cell lymphoma, which is often refractory to treatment and carries a poor prognosis. Most RS cases are clonally related to the pre-existing CLL, with *TP53* disruption being one of the most frequent underlying genomic alterations[25,26]. Clinically, the diagnosis of RS is often not straightforward and relies on a high index of clinical suspicion and accurate tissue biopsy. Prompt recognition and treatment of RS is important; however, currently there are no

strategies that allow for early diagnosis or serial monitoring of molecular changes that may predict impending transformation. In our cohort, three individuals (CLL004, CLL054 and CLL022) developed RS following treatment (Fig. 2 and Supplementary Figs 5–7). To evaluate whether plasma analysis could detect the molecular events underpinning disease transformation, we performed both low-coverage whole-genome sequencing (LC-WGS) on paired plasma (P) samples and whole-exome sequencing (WES) on paired PB, BM or LN tissue, plasma and MNL DNA, at baseline and RS in all three cases.

In CLL004 and CLL054, RS developed within the LN compartment following therapy with venetoclax and ibrutinib, respectively. BM analysis of both cases at the time of RS showed no evidence of transformation, suggesting compartmentalized

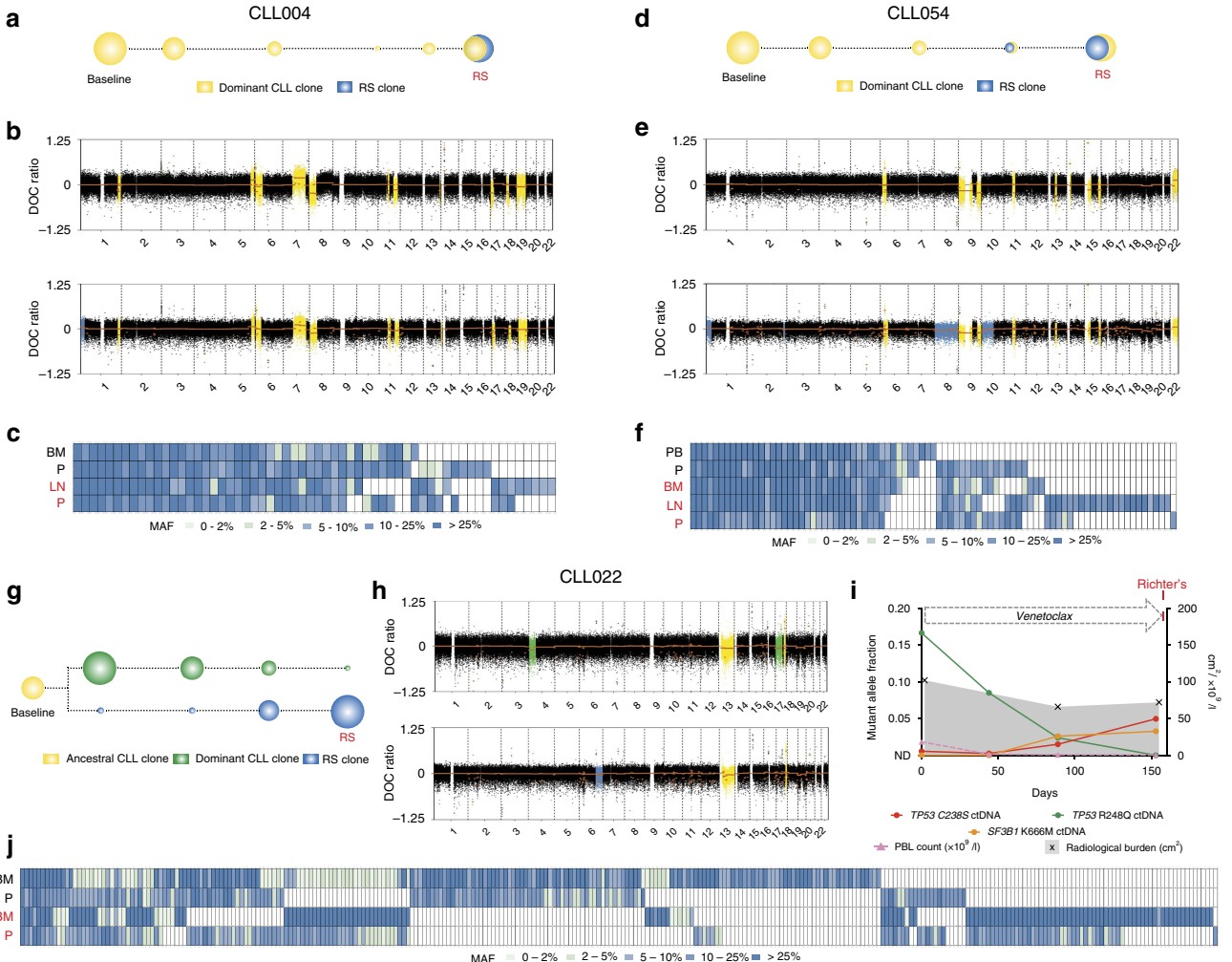

**Figure 2 | ctDNA detects clonal evolution in CLL patients with RS.** (**a**) Graphical representation of the linear pattern of evolution of CLL to RS in CLL004 revealed by LC-WGS of the plasma and WES of plasma (P) and BM/LN. (**b**) Depth of coverage (DOC) log$_2$ ratio plots from LC-WGS of plasma in CLL004 at baseline (top panel) and at RS (bottom panel) showing CNAs specific to the dominant CLL clone in yellow and new CNAs at the time of RS in blue. (**c**) Heat-map illustrating the distribution of predicted functional SNVs from WES at baseline (BM and P) and at progression to RS (LN and P shown in red) in CLL004. (**d**) Graphical representation of the linear pattern of evolution of CLL to RS in CLL054. (**e**) DOC log$_2$ ratio plots from LC-WGS of plasma in patient CLL054 at baseline (top) and at RS (bottom) showing CNAs specific to the dominant CLL clone (yellow) and new CNAs at the time of RS (blue). (**f**) Heat-map illustrating the distribution of predicted functional SNVs from WES at baseline (PB and P) and at progression to RS (BM, LN and P shown in red) in CLL054. (**g**) Graphical representation of the branched pattern of evolution of CLL to RS in CLL022. (**h**) DOC log$_2$ ratio plots from LC-WGS of plasma in CLL022 at baseline (top) and at RS (bottom) showing CNAs specific to the ancestral CLL clone (yellow), the dominant CLL clone (green) and new CNAs at the time of RS (blue). (**i**) ctDNA levels in CLL022 showed a decrease in the *TP53* p.R248Q mutation post venetoclax treatment. In contrast, there was an increasing fractional abundance of *TP53* p.C238S and *SF3B1* p.K666M mutations observed in the plasma as the patient progressed to RS. These mutations were detected in plasma 64 days before the clinical diagnosis of RS. (**j**) Heat-map illustrating the distribution of predicted functional SNVs from WES at baseline (BM and P) and at progression to RS (BM and P shown in red) in CLL022.

progression at the nodal site. Genomic analyses in these two cases revealed that the RS had arisen from the dominant CLL clone (Fig. 2a–f). Tumour tissue WES showed shared copy number alterations (CNAs) at baseline and at RS, with the acquisition of new RS-specific CNAs identified at the time of disease progression (Supplementary Fig. 6). Importantly, these CNAs were readily detectable in the plasma by LC-WGS including identification of the known 17p deletion in CLL004 (Fig. 2b,e). At the time of RS, the CNAs identified in plasma were a mixture of those specific to the CLL disease (within the BM) and those new to the RS (within the LN), highlighting the ability of ctDNA analysis to comprehensively represent genomic changes across multiple disease sites. In keeping with the CNA analysis, tumour tissue WES also revealed a high proportion of shared

single-nucleotide variants (SNVs) between the baseline and RS samples (CLL004; 37/55 (67%) and CLL054; 35/83 (42%)) with several new SNVs emerging at the time of RS (Supplementary Data 1 and 2). In each case, many of the shared and RS-specific SNVs were identified in the plasma and several of the RS-specific SNVs were also seen in plasma at baseline (Fig. 2c,f). Importantly, the RS-specific SNVs could only be readily identified in the plasma and tumour tissue analysis, but were not identified in the MNL DNA at the time of transformation (Supplementary Figs 5 and 6).

In the final case, CLL022, the RS developed within the BM compartment after venetoclax therapy and genomic analysis revealed a different pattern of evolution whereby the RS had arisen from transformation of an ancestral CLL clone (Fig. 2g–j).

Plasma LC-WGS showed a dramatic change in the plasma CNA profile at RS compared with baseline, including a del(17p) at baseline, which was not present in the RS sample (Fig. 2h). These findings were confirmed through tumour tissue WES and *TP53* fluorescence *in situ* hybridization (FISH; Supplementary Fig. 7). In support of the branched pattern of evolution, tumour tissue WES revealed only 84/282 (30%) shared SNVs between the baseline and RS samples, with a large number of CLL-specific (128/282 (45%)) and RS-specific (70/282 (25%)) SNVs identified (Fig. 2j and Supplementary Data 3). Importantly, the plasma closely reflected these changes both at baseline and at the time of RS (Fig. 2j). In contrast, analysis of the MNL DNA could not identify the RS-specific changes at the time of transformation (Supplementary Fig. 7). Notably, the analysis revealed two *TP53* mutations (one mutation at baseline and a separate mutation at RS), as well as a *SF3B1* mutation detected at RS (Supplementary Data 3). TS and dPCR data for these mutations across four serial plasma samples showed a decline of the initial *TP53* mutation associated with treatment response of the dominant CLL clone, followed by emergence of the new *TP53* and *SF3B1* mutations ~2 months before the clinical diagnosis of RS (Fig. 2i and Supplementary Fig. 8). Our analysis of the three RS cases did not reveal recurrent genomic alterations that were common across all cases. However, these data highlight the ability of ctDNA analysis to reflect clonal evolution under the selective pressure of therapy and provide a useful strategy to facilitate early identification of treatment failure in CLL.

## Discussion

ctDNA is an emerging biomarker in the management of solid cancers and non-Hodgkin's lymphoma[27–29]; however, its role in haematological malignancies with a circulating disease component has not been previously explored. Here we provide the first report describing the detection of ctDNA in patients with relapsed/refractory CLL and demonstrate the potential of this approach for molecular disease monitoring by providing a methodology that (i) can comprehensively capture the underlying genomic profile, (ii) can monitor aggregate disease burden integrating different topographical sites within the body and (iii) can allow the discovery of new genomic changes associated with resistance or clonal advantage over time. Although current methods of disease monitoring such as multicolor flow cytometry allow assessment of disease burden with high sensitivity, these existing approaches do not provide critical insights into the genomic evolution of the disease. Our findings support the introduction of ctDNA analysis into future CLL clinical trials as a complementary strategy to existing methods of assessing disease status. Together, these combined modalities will lead to a comprehensive assessment of tumour burden across multiple disease sites and help define the molecular events that underpin response and failure to current and emerging therapies in this disease.

## Methods

**Patients and sample collection.** The cohort of 32 patients described in this study were enrolled in an ongoing prospective clinical study for the collection of biospecimens in patients undergoing treatment with novel therapeutics for relapsed CLL. The study was approved by The Peter MacCallum Cancer Centre research ethics committee (13/36). All blood and tissue samples collected and utilized for the purpose of this study were obtained with written informed consent from each patient.

**Sample processing and DNA extraction.** Blood was collected in ACD (Sarstedt SMonovettes 8.5 ml CPDA) tubes and was processed within 1 h of collection. Whole blood was first centrifuged at 1,600 *g* for 10 min (brake off) to separate the plasma from the PB cells, followed by a further centrifugation step at 20,000 *g* for 10 min to pellet any remaining cells and/or debris. The plasma was

then stored at −80 °C until DNA extraction. DNA was extracted from 2 ml aliquots of plasma using the QIAamp Circulating Nucleic Acid Kit (Qiagen, 55114) according to the manufacturer's instructions. The DNA was then eluted into 50 μl buffer AVE (Qiagen) and stored at −20 °C. To estimate the efficiency of DNA extraction, equal amounts of a diluted DNA amplicon (PCR amplified from *Drosophila melanogaster* genomic DNA, Supplementary Table 4) was spiked into each plasma sample before extraction. The amount of spike-in *Drosophila* DNA was then quantified in each eluted plasma DNA sample by dPCR and the efficiency of extraction was determined by comparing this quantity with a no-extraction control that was prepared by re-suspension of the same amount of *Drosophila* DNA in the same elution medium.

After the collection of plasma from each blood sample, the mononuclear cell layer (MNL) was collected using the Ficoll-Paque Plus (GE Healthcare, 17144002) separation method, according to the manufacturer's protocols. Red cell lysis was performed using a red cell lysis buffer (155 mM NH$_4$Cl, 10 mM KHCO$_3$ and 0.1 mM EDTA pH 7.4). MNL cells were then enumerated using a haemocytometer and aliquots of $5 \times 10^6$ cells were snap-frozen in liquid nitrogen and stored at −80 °C. BM aspirates were similarly processed by the Ficoll separation method to obtain bone marrow-derived cells. DNA was extracted from the BM or PB MNL cells using the DNA Blood and Tissue kit (Qiagen, 69504), using the manufacturer's protocols. DNA was extracted from formalin-fixed, paraffin-embedded LN biopsy sections using the QIA-amp DNA FFPE Tissue Kit (Qiagen, 56404), according to the manufacturer's protocols.

Matched normal DNA was obtained from all study participants to enable confirmation of somatic status of identified genetic events. For CLL018 and CLL020, Omni swabs (GE Healthcare, WB100035) were used to wipe the buccal lining ~10 s. Swabs were then processed for DNA extraction using the DNA Investigator Kit (Qiagen, 56504). For all other cases, ~2 ml of saliva was collected in an Oragene Saliva Collection Kit (DNA Genotek, OG500). DNA was then extracted from saliva using the prepIT L2P kit (DNA Genotek, PTL2P45), according to the manufacturer's protocols.

**Digital PCR.** dPCR analysis was performed using the BioMark 48.770 Digital Array (Fluidigm) system following the manufacturer's protocols. Allele-specific PCR assays to specifically detect and quantify the fractional abundance of point mutations and corresponding wild-type alleles were either custom designed or commercially obtained (PrimePCR PCR Primers and Assays, BioRad Laboratories) (Supplementary Table 4). The total amount of cell-free DNA was quantified using an assay that targets a non-amplified region in the genome, the *RPP30* gene on chromosome 10, as described previously[30]. Each sample was analysed by at least two technical replicates comprising of at least 1,540 individual reactions. A Poisson correction was applied to determine the number of amplifiable molecules, which was used to further derive the number of copies of DNA carrying a particular mutation per millilitre of plasma. Data analysis was carried out using the Digital PCR Analysis Software version 4.1.2 (Fluidigm).

**Targeted sequencing.** TS was performed using the 48.48 Access Array system (Fluidigm), as described previously[30]. A panel of 51 amplicons with an average size of 170 bp was designed across known COSMIC mutations in seven genes recurrently mutated in CLL, that is, *SF3B1*, *NOTCH1*, *ATM*, *TP53*, *MYD88*, *KRAS* and *BIRC3* (Supplementary Table 5). Plasma and MNL DNA across several time points with corresponding matched normal DNA was then amplified with tagged target-specific primers using the microfluidic platform, which allowed 48 samples to be analysed simultaneously with multiplexed assays in 2,448 reaction chambers. DNA derived from plasma and buccal swab samples were first subjected to pre-amplification in a 10 μl reaction volume containing a pool of forward and reverse target-specific primers using the Fast Start High Fidelity PCR System (Roche, 04738292001). The pre-amplified samples were diluted fivefold in PCR-grade water before target-specific amplification using the Access Array system. DNA from MNL cells and saliva samples was not pre-amplified and 50 ng DNA was used directly for target specific amplification using the Access Array system. Following amplification, products were harvested, tagged with sample-specific barcodes, pooled together and purified using AMPure XP beads. All samples were analysed in duplicate to control for PCR artefacts. The purified libraries were then sequenced using 150 bp read length on the Illumina MiSeq sequencer. Sequenced reads were mapped to the human reference genome (version hg19) using BWA-MEM (version 0.7.12) with default parameters. The mean targeted sequencing coverage was 2,171-fold for plasma and 8,789-fold for MNL and matched normal DNA. Mutations with a minimum of five reads supporting the variant and an MAF >1% were retained for further analysis. Variants that were recurrently observed in >50% of the samples (representing probable sequencing and/or PCR artefacts) and those with a high global allele frequency (>1.0%) in the 1000 genomes database were flagged and removed from this curated list. Variants in the curated list were then annotated based on their prognostic or functional relevance, as described previously[31].

**IGH dPCR analysis.** IGH sequencing of baseline PB MNL samples was performed for CLL001, CLL018, CLL020 and CLL043 by next-generation sequencing using the BIOMED-2 FR1 primers[32] containing the Fluidigm universal forward and

reverse sequencing tags (CS1 and CS2). Approximately 200 ng genomic DNA was PCR amplified using the Roche FastStart High Fidelity PCR System. PCR conditions were an initial denaturation step of 95 °C for 2 min followed by 35 cycles of 95 °C for 45 s, 60 °C for 45 s and 72 °C for 90 s, and a final elongation step of 72 °C for 10 min. The resulting PCR products were then used as template in a second PCR reaction with sample-specific barcode primers (Fluidigm) as per the manufacturer's instructions. Uniquely indexed samples were pooled and the resulting library was purified using the Agencourt AMPure XP system (Beckman Coulter). The resultant library was quantified on a Tapestation 2200 (Agilent Technologies). Libraries were denatured and diluted as per the manufacturer's instructions and 300 bp paired-end sequencing was performed on the Illumina MiSeq sequencer. Paired-end reads were assembled using PEAR (v0.9.8)[33] and analysed with an in-house-designed interactive web-based tool displaying the sequences and read-length distribution of assembled reads. Dominant IGH sequence was identified for each case (Supplementary Table 6), for subsequent dPCR assay design. Patient-specific IGH dPCR assays were then designed and analysed on the BioMark 48.770 Digital Array (Fluidigm) platform using DNA binding dye (Evagreen) (Supplementary Table 4) as described earlier.

**Exome sequencing.** WES sequencing was performed on tissue and plasma samples from three CLL patients who underwent Richter's transformation (Supplementary Data 1–3). Saliva DNA was used as a germline control. Tissue and saliva DNA were sheared by sonication (Covaris). Exome enrichment was performed using the NimbleGen SeqCap EZ Exome Kit v3.0, following the manufacturer's protocols. Plasma DNA samples were sequenced in two rounds for samples from CLL004 and CLL022, to achieve desired depth of coverage. Indexed pools of three to four samples per lane were sequenced on the Illumina HiSeq2000 or Illumina NextSeq500 instrument. Mean total depth of coverage was ∼60× for germline samples, ∼140× for BM and LN-derived DNA samples, and ∼120× for plasma samples. Detailed sequencing metrics have been provided in Supplementary Data 4.

FASTQ sequences files for each sample were evaluated with FASTQC (version 0.110.3) for quality control and Cutadapt (version 1.8.1) was used to remove adapter sequences, primers and other sequence artefacts. The reads were then mapped to the human reference genome (version hg19) using BWA-MEM (version 0.7.12) with default parameters. PCR duplicates were removed using Picard (version 1.133). Local realignment was performed using GATK (version 3.4.0) IndelRealigner. GATK's BaseRecalibrator was then used for recalibration of base qualities. SAMtools mpileup command was used to create pileup files for variant calling. SNV and indel detection was performed with GATK, VarScan (version 2.3.8), MuTect (version 1.1.7) and multiSNV[34]. Mutations were annotated for read depth of reference and alternate alleles, and Ensembl Variant Effect Predictor (version 80) and COSMIC (version 72) were used to explore the impact of the detected somatic mutations. We retained predicted functional mutations (nonsense, missense or splicing SNVs) for further analysis. All variants with mutant reads observed in the matched normal sample were discarded. Mutations that had strand bias and those detected within homopolymer regions were also removed.

High-confidence SNV and indel calls were defined by those variants predicted by at least two variant callers (GATK, VarScan and/or MuTect)[35]. If a high-confidence variant was observed in a sample(s), but not in other samples from the same individual, we applied an additional variant algorithm, multiSNV[34], which uses a Gibbs Sampler to detect variants in multiple related samples. If the variant was detected in the related samples using multiSNV, it was also reported as high confidence. Variants supported by at least two alternate reads, and a minimum total read depth of 20, but which did not meet the above criteria are denoted by * (Supplementary Data 1–3). These variants had been identified as high-confidence variants in at least one other tumour or plasma sample from the same individual. To adjust for purity and ploidy, we used the average of purity estimates from ADTEx[36] and Sequenza[37], the copy number estimates generated by Sequenza and adjusted MAF using the formula described by Nik-Zainal et. al.[38]

ADTEx was used for whole exome copy number analysis, using a hidden Markov model that determines copy number changes by adjusting for and estimating B-allele frequency (BAF), ploidy and sample purity[36].

**Low-coverage whole-genome sequencing.** LC-WGS was performed on plasma samples for CLL004, CLL052 and CLL022 at baseline and at time of RS. Libraries were prepared and sequenced on the Illumina NextSeq500 platform (paired-end 75 bp) according to standard protocol[39]. Detailed sequencing metrics have been provided in Supplementary Data 4. Copy number analysis was performed using QDNASeq. The genome was divided into non-overlapping 15 kb bins and other parameters were set as default as per QDNASeq. The number of reads in each bin is counted and adjusted with a simultaneous two-dimensional loess correction accounting for read mappability and GC content[40].

**Clinical assessment of disease.** Lymphadenopathy was measured by direct clinical assessment by the treating haematologist for CLL012 and CLL043, magnetic resonance imaging for CLL008, computed tomography (CT) and

positron emission tomography for CLL022, and by serial CT scans, where available, for all other cases. For patients with measurable disease, a blind review by an experienced radiologist of all CT/magnetic resonance imaging images was conducted to document objective response or progressive disease according to the International Workshop on CLL (iwCLL) criteria[18,41]. Radiological disease burden (expressed in $cm^2$) was determined by the sum of the product of the perpendicular diameters of up to six target lesions imaged along the treatment timeline. For patient CLL012, clinical LN burden was calculated by the sum of the product of the perpendicular diameters of six target lesions as measured by the treating haematologist.

Correlation analysis was performed at timepoints where matched radiological disease burden, PBL count, plasma MAF and MNL DNA MAF were assessed within a 2-week window (Fig. 1d and Supplementary Table 2) ($n = 38$ time points from 12 patients). The change in each parameter at each time point was calculated as a percentage of the maximal value observed across all time points in each case. The normal reference range for PBL counts was $1–4 \times 10^9$/l.

Four-micrometre sections of B5-fixed BM biopsies (or formalin-fixed LN biopsy) were stained with haematoxylin and eosin to assess histology using standard diagnostic laboratory protocols. FISH analysis was performed on case CLL022, where BM samples were cultured, fixed and harvested according to standard cytogenetic techniques (Victorian Cancer Cytogenetics Service, St Vincent's Hospital, Melbourne, Victoria). FISH analysis was performed using the TP53/ATM dual-colour probe (Cytocell Aquarius, Cambridge UK). Slides and probe underwent co-denaturation at 75 °C for 2 min followed by overnight hybridization at 37 °C. After washing (using the Vysis rapid wash protocol), the slides were air dried and mounted with 4,6-diamidino-2-phenylindole/anti-fade.

**Data availability.** The sequencing data that support the findings of this study have been deposited into the sequence read archive, which is hosted by the National Centre for Biotechnology Information. The BioProject accession number is PRJNA361191.

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

## Acknowledgements

P.Y. is supported by the Klempfner Epigenetics Fellowship administered by the Snowdome Foundation and by the Haematology Society of Australia and New Zealand new investigator scholarship. A National Breast Cancer Foundation and Victorian Cancer Agency Fellowship currently support S.J.D. The National Health and Medical Research Council of Australia (1104549) supported this research. The clinical study patient recruitment was funded by Abbvie and Janssen. We thank Sreeja Gadipally and Timothy Holloway for assistance with exome sequencing and LC-WGS.

## Author contributions

S.J.D. and C.T. designed the project. C.T. and J.S. recruited participants. C.T., J.S., D.S., P.Y., P.P., D.W. and P.B. provided patient samples and clinicopathological data. E.W., D.S., S.F., P.Y., R.A. and A.Z. developed the targeted gene panel with helpful input from P.B. and D.W. D.S., S.F., P.Y., S.W., R.V., E.W., M.J.S., G.M.A., T.S. and K.J. performed the experimental work. D.S., P.Y. and T.H. analysed data with input from A.P., K.D., Y.C.C. and E.L. D.S., P.Y., T.H., M.D., C.T. and S.J.D. interpreted the data. M.W. analysed cytogenetic and FISH data. D.J. performed analysis of radiological measurement. D.S., P.Y., T.H., M.D., C.T. and S.J.D. wrote the manuscript. All authors approved the final version of the manuscript.

## Additional information

**Competing financial interests:** The authors declare no competing financial interests.

**Publisher's note**: 

