## [Peer Review File · Nature Communications]

Reviewers' comments:

Reviewer #1 (Remarks to the Author):

The authors have substantially increased the cohort of patients they have studied, and this reviewer finds that the authors have satisfactorily addressed the critiques of the original manuscript.

Reviewer #3 (Remarks to the Author):

In response to my question why ctDNA would be a valuable alternative to established testing methods of CLL cells from blood or marrow the authors respond as follows:

“Whilst CLL patients have leukemia cells "readily accessible in blood", it is increasingly clear that the circulating CLL cells are not representative of the overall disease burden especially in the context of novel therapies. The disease contained within the marrow and lymph nodes are the most proliferative sites, yet these are not readily accessible and require invasive biopsy for molecular assessment. Serial tissue biopsies at frequent intervals throughout treatment are not feasible in most patients. In comparison serial analysis of ctDNA is minimally invasive, can be repeated regularly and provides a molecular approach for monitoring the tissue compartment without the need for regular biopsies. ctDNA is released directly from cells within the tissue compartment and provides different molecular information to that provided from analyses of the circulating leukemic cells. This is exemplified in several cases across our cohort with evidence of disease compartmentalization (Figure 1 and 2). In CLL, the bone marrow and lymph node compartments represent critical sites of active disease and the ability to monitor these compartments, in addition to the circulating leukemic cells, using a minimally invasive molecular test, has the same key advantages as that seen in solid malignancies.”

This response is not to the point and therefore does not well address my concerns. The authors talk in the first sentence about disease burden, which is a question that probably cannot be answered by this technology, at least with the limited provided data. Disease burden is the sum of blood and tissue involvement, and can only be estimated by analyzing marrow involvement, blood involvement, and lymph nodes and other tissues. The authors have not attempted to estimate overall disease burden or changes thereof. Instead they look at decline in blood ctDNA levels that is paralleled by a decline in radiographic signs of disease, which is a very crude way of comparing lymph node to ctDNA levels. Then they argue that circulating CLL cells are not representative of the disease, especially with novel therapies. The authors do not provide any data or reference to support such statement. The current disease pathogenesis model favors a model of CLL cell recirculation between tissue and blood, which would argue that the blood compartment provides a representative sample of the entire disease. This is not mentioned or discussed. In the next sentence the authors accurately state that CLL cells proliferate in the tissues and not the blood. But whether the ctDNA testing is more suitable for picking up any proliferation-related signals than established blood or marrow tests is not clear. Proliferation goes along with transcriptional activation, but the ctDNA testing would not pick up such events. What is the relation between ctDNA and proliferation? The authors continue stating that ctDNA is released directly in tissues, what is the evidence for that? I think, overall, the authors need to be more precise and less advertising about what exactly we can learn from ctDNA testing in CLL, based on the current data. ctDNA can be tested from PB and probably contains ctDNA from CLL cells in all disease compartments. ctDNA levels may reflect overall disease burden, but that would need to be better

worked out using, for example, models to estimate overall disease burden by integrating volumetric assessment of tissue disease, the blood compartment, as well as the marrow compartment. ctDNA further can be useful in cases where the disease mostly resides in the tissue, like in SLL, or where therapy has eliminated blood but not tissue disease. A presentation and discussion of the data in such context would make more sense to me.

To state that ctDNA testing is minimally invasive is too advertising; any other blood testing which can give us valuable information is equally minimally invasive; I suggest to remove "minimally invasive".

The authors answer my question of ctDNA testing versus more traditional methods of disease monitoring by stating "In contrast, total tumor burden assessment by ctDNA accurately reflected the decrease in CLL burden, and was capable of "looking past" the artificial rise in white cell count" when talking about ibrutinib therapy and transient lymphocytosis. This statement is based on declining ctDNA levels during ibrutinib treatment. Following that argument one could say that cytokines which rapidly decline after start of ibrutinib therapy also reflect disease burden, and measuring such cytokines would be more simple than ctDNA testing. But the main issue with this statement is that the authors have not formally looked at disease burden and do not prove that ctDNA levels reflect tissue disease burden. First, because the lymphocytosis in ibrutinib-treated patients is not artificial, it is very real and it may represent all or only a fraction of the tissue disease which is being shifted from the tissues into the blood. If all tissue CLL cells are mobilized into the blood, then declining ctDNA levels would be an inaccurate representation of overall disease burden. Second, the model therefore lacks an integrated approach of connecting blood, marrow, and tissue compartment CLL burden with the ctDNA data. Either add such model data, or tone down the statements and interpretation of the data, as suggested above.

The authors state on page 4 that ctDNA levels declined during ibrutinib therapy, unless there was an increase in disease burden radiographically. That would be highly unusual, CLL patients progress on ibrutinib at early time points only if they have Richter transformation. In how many patients was the latter seen? Were these Richter cases?

Minor:

On page 3 the PI3K inhibitors should also be mentioned, given that idelalisib is FDA approved.

Reviewer #4 (Remarks to the Author):

Sinha et al report an interesting paper describing the use of circulating tumor DNA (ctDNA) in CLL. Circulating tumor DNA is an important and emerging topic, with significant potential clinical utility. ctDNA has not been studied in this specific disease before, but has been studied in other B cell malignancies, such as non-Hodgkin lymphomas. The authors use a number of methods to describe the biology of ctDNA, including targeted NGS, digital PCR, and WES/ low depth WGS in 3 patients with Richter's syndrome.

While provocative, there are a number of points requiring further clarification, as well as limitations to the manuscript in its current form. Specifically, the authors make two major claims in their paper:

- 1) That ctDNA is more sensitive for detection of disease than MNL DNA (confirming prior results in NHL from Johns Hopkins, Stanford, and NCI that should be cited [PMIDs 21399237, 25887775, 25842160]), and
- 2) that ctDNA reflects the genomic evolution of CLL into Richter's Syndrome better than alternative methods.

Regarding claim 1):

One of the strengths is the specific focus on CLL and the number of patients and samples for this specific assessment. This does lend more credence to the claim that tumor DNA is more detectable

as ctDNA than in MNLs. However, this comparison was made across many different time-points, including both pretreatment and after administration of Rituximab or Obinituzumab, which are targeted CD20 antibodies. These agents are expected to rapidly remove any leukemia cells from circulation, and therefore, at these timepoints, it is less surprising that ctDNA is superior to MNL DNA for detection of disease. The authors should clarify which timepoints were used for the samples for comparison between ctDNA and MNL (ie: how many samples were from before therapy and how many were after therapy, and was the difference between ctDNA and MNL DNA seen at all times?)

Additionally, the authors attempt to demonstrate that ctDNA better reflects radiographic disease burden than MNL DNA (figure 1d). This is an important point. However, it is not clear how the change in radiographic burden was calculated (ie: which timepoints), and how many of the points represent unique patients versus patients with multiple scans and multiple ctDNA measurements (that is, does this n=38 mean 38 individual patients?). There is a reference to the "maximal value" or radiographic disease burden in the legend – how is this determined? Is this from pretreatment CT or PET/CT, and were the quantitative measures calculated using established criteria (eg, Cheson criteria) as relevant for trial endpoints? A more detailed description of the methods is needed, and an accompanying table in the supplement of the raw data would be useful. Separately, seeing that response criteria in CLL typically do not primarily rely on radiographic assessments, the authors should relate their findings in assessing the fidelity of ctDNA vs MNL as correlates of more traditional CLL response criteria.

Regarding claim 2):

The authors also claim that ctDNA can reflect clonal evolution in CLL, and provide evidence from 3 cases that support this claim, examining emerging copy number alterations and emerging SNVs. This is interesting and compelling data, however, this conclusion is based on a low number of cases (3 patients with Richter's Syndrome of 25 total patients). The generalizability of this finding is limited, given the low number of cases. Furthermore, there does not seem to be any overlying new biological finding here, other than that novel mutations can be seen in the plasma. More detailed information should be provided as to what these novel mutations are. Was any pattern of emerging mutations seen?

The authors also frame their introduction around the importance of novel agents in CLL, such as Ibrutinib and Venetoclax. In fact, all of the patients in this study received one of these drugs. However, there is no attempt to link their findings from WGS and WES in the three patients with Richter's Syndrome back to their novel therapy. No major conclusions are possible regarding the biology of CLL after treatment with Ibrutinib and Venetoclax, which is potentially due to the low number of patients (n=3) and anecdotal nature of this data. Therefore, the biological novelty of the study is limited.

Ibrutinib and Venetoclax are both agents approved in CLL patients harboring 17p deletions spanning TP53, which is frequently a target of deletion during clonal evolution and clonal selection of CLL during therapy. It appears that this lesion was present nearly all of the patients profiled by the authors, ostensibly as detected by FISH on tumor-rich specimens as the basis for eligibility. The authors should comment on the noninvasive genotyping of 17p deletions using ctDNA in this same cohort, whether by targeted amplicon sequencing, WES, and/or low pass WGS.

Furthermore, and perhaps more importantly, the authors do not show WES data from the mononuclear cells (MNL) at the time of Richter's Syndrome transformation. As the authors state that ctDNA, and not MNL DNA, accurately represents the whole body experience of disease, this seems like an important control (ie: do the emergent mutations that are seen in Richter's appear in the MNL?) The authors should include WES from the mononuclear cells at these timepoints.

Finally, the authors demonstrate two forms of clonal evolution at the time of Richter's Syndrome – linear evolution (CLL004 and CLL054) and branched evolution (CLL022). Are these types of

evolution specific to Richter's Syndrome, or can they be seen in relapsing CLL without RS? There is no comparison to relapsed CLL without Richter's Syndrome. Inclusion of patients with relapsing CLL without RS in the WGS/WES analysis presented in Figure 2 would add to the biological insight of this work, if such data is available.

Additional minor comments:

- Line 80-82 – "somatic driver mutations were identified in 25/32 individuals in both matched MNL and plasma DNA samples" – Table S1 does not contain the information as to which fraction the mutation was identified in, or the AF. It would be useful to know if the driver mutations were identified in MNL, plasma, or both.

- Line 129-133 – (Figure 1 C) "Case CLL001 provides an example of this scenario, where despite evidence of a significant treatment response to obinutuzumab in PB and BM, the patient developed progressive lymphadenopathy. This was paralleled by a sharp rise in ctDNA highlighting the ability of ctDNA to reflect nodal disease progression following treatment."

I am not sure that this figure supports this statement. The patient does appear to have a large spike in ctDNA from ~day 100 to ~ day 300 (ctDNA AF increases ~10x), however, the nodal disease burden (grey area plot) does not appear to increase nearly as much (increase from ~12mm² to 15mm² of nodal disease). The authors claim that the spike in ctDNA represents this change in radiological disease burden, but these seem to be on different orders of magnitude. Could there possibly be an alternative biological reason for the increase in ctDNA? This statement seems too strong as is.

- Figure 2b and 2e (copy number plots from WGS of plasma at time of Richter's) – these plots are attempting to show novel copy number alterations at the time of Richter's transformation from low coverage WGS of the plasma (areas shown in red). However, the copy number plots themselves are not impressive, as the red areas do not by eye appear to be copy number altered compared to the black areas. In the methods, the authors state that copy number analysis was performed using QDNaseq developed for tumor specimens. The performance of this method on ctDNA has not been established, especially in terms of detection limits and Sn/Sp. More details as to how copy number alterations were actually called, thresholds used, etc should be provided.

- Regarding the clonal evolution of CLL during Richter's Syndrome, the authors state that "In ... tumor tissue WES also revealed a high proportion of shared single nucleotide variants (SNVs) between the baseline and RS samples ... with several new SNVs emerging at the time of RS ... In each case, many of the shared and RS-specific SNVs were also identified in plasma, and several of the RS-specific SNVs were also seen in plasma at baseline (Fig. 2c and 2f)". However, the heatmaps shown in 2c and 2f do not have mutation or gene labels. These should be added to the heatmaps (likely with the heatmaps rotated 90 degrees), so that the emergent mutations can be more specifically seen.

- There are only 25 cases in Table S1, but in the supplementary methods section the authors mention 32 patients. The 32 patients likely include patients in who targeted sequencing did not reveal mutations. Was there anything special about these 7 patients, as opposed to the 25 where mutations were seen? The clinical information for these patients should be made available.

- Figure 2 describes multiple subclones in patients with CLL and emerging clones in Richter's Syndrome. Are the sizes of the circles in figure 2a, 2d, and 2g meant to represent the sizes of each clone? If so, how were the sizes of these circles determined? Also, it appears that the heatmaps of mutations in 2c, 2f, and 2j are clustered. How was this clustering performed?

Reviewer #1 (Remarks to the Author):

The authors have substantially increased the cohort of patients they have studied, and this reviewer finds that the authors have satisfactorily addressed the critiques of the original manuscript.

We thank the referee for their support and suggestions, which have improved our manuscript.

Reviewer #3 (Remarks to the Author):

In response to my question why ctDNA would be a valuable alternative to established testing methods of CLL cells from blood or marrow the authors respond as follows:

“Whilst CLL patients have leukemia cells "readily accessible in blood", it is increasingly clear that the circulating CLL cells are not representative of the overall disease burden especially in the context of novel therapies. The disease contained within the marrow and lymph nodes are the most proliferative sites, yet these are not readily accessible and require invasive biopsy for molecular assessment. Serial tissue biopsies at frequent intervals throughout treatment are not feasible in most patients. In comparison serial analysis of ctDNA is minimally invasive, can be repeated regularly and provides a molecular approach for monitoring the tissue compartment without the need for regular biopsies. ctDNA is released directly from cells within the tissue compartment and provides different molecular information to that provided from analyses of the circulating leukemic cells. This is exemplified in several cases across our cohort with evidence of disease compartmentalization (Figure 1 and 2). In CLL, the bone marrow and lymph node compartments represent critical sites of active disease and the ability to monitor these compartments, in addition to the circulating leukemic cells, using a minimally invasive molecular test, has the same key advantages as that seen in solid malignancies.”

This response is not to the point and therefore does not well address my concerns.

We apologise if we misunderstood the referee's original question, which specifically stated "ctDNA monitoring has great potential in solid cancers, where primary tumors and/or metastases often are not accessible for testing. In contrast, CLL patients have leukemia cells readily accessible in blood, marrow, and lymph nodes that can be isolated for disease monitoring, including monitoring for clonal changes and emerging resistant subclones. Therefore, the advantage of using ctDNA testing in CLL is not obvious". We concluded from this statement that the referee clearly appreciates the potential of ctDNA monitoring in solid malignancies but was uncertain of its role in a malignancy where the tumor cells are circulating. Therefore, our response was specifically written to address the question of what can be gleaned from ctDNA analysis that cannot be obtained from accessible tumor cells within blood.

Firstly, it is important to recognize that bone marrow and in particular lymph node biopsies are no less invasive than the tissue biopsies needed to assess the cancer genome in solid malignancies. If the referee sees the 'great potential' of ctDNA in solid malignancies where tissue is not easily accessible, that same value and potential applies in this setting.

Secondly, if the genomic information gained from ctDNA simply mirrored that obtained from the circulating malignant cells we would also agree that there is no added value of ctDNA in this malignancy. However, this is clearly not the case and there are numerous examples provided throughout the manuscript where ctDNA provides genomic information of clinical relevance that was not apparent in the circulating malignant lymphocytes. The patients who have compartmentalized disease progression and/or disease transformation best exemplify this fact (Fig. 1c, Fig. 1e, Fig. 2).

The authors talk in the first sentence about disease burden, which is a question that probably cannot be answered by this technology, at least with the limited provided data. Disease burden is the sum of blood and tissue involvement, and can only be estimated by analyzing marrow involvement, blood involvement, and lymph nodes and other tissues. The authors have not attempted to estimate overall disease burden or changes thereof. Instead they look at decline in blood ctDNA levels that is paralleled by a decline in radiographic signs of disease, which is a very crude way of comparing lymph node to ctDNA levels.

We are not aware of any method to assess overall disease burden in CLL (or indeed in any haematological malignancy) that collectively analyzes bone marrow, lymph node and blood involvement simultaneously. If the referee is aware of a method or model to assess overall disease burden in this manner, then we would readily apply this methodology and compare it to ctDNA. Instead we have employed the current gold standard methodology, which is utilised in clinical practice to assess disease burden in CLL (i.e. iwCLL criteria). Whilst we agree with the referee that radiographic imaging is a crude assessment of lymph node disease, this remains the standard to which all clinicians practice according to the iwCLL criteria and therefore we believe it is the most relevant metric to compare with ctDNA. It is clinically important and relevant to provide this comparison to the readers. However, we have also revised the manuscript to remove the emphasis on "overall disease burden" and instead we now focus on the comparison of ctDNA with each test currently used to clinically assess disease burden. For instance we now specifically state that changes in ctDNA levels more accurately reflect changes in radiological disease burden than peripheral blood MNL DNA.

Then they argue that circulating CLL cells are not representative of the disease, especially with novel therapies. The authors do not provide any data or reference to support such statement.

We apologise if this statement was not clear. What we are trying to reinforce here is that the peripheral blood lymphocyte count which is currently used as a

surrogate of disease response and/or activity in CLL is not reliable in the context of novel therapies and we have provided data to support this statement. With ibrutinib the marked drug associated lymphocytosis makes monitoring of peripheral blood counts to assess disease response obsolete (Figures 1b and S4). Similarly, the marked lymphopenia observed with Venetoclax makes monitoring of peripheral blood counts to measure disease activity unreliable (Figures 1a and S3). Furthermore, we have shown that compartmentalised disease progression and/or transformation on novel therapies can occur without an ensuing lymphocytosis (Figures 1C and 2).

The current disease pathogenesis model favors a model of CLL cell recirculation between tissue and blood, which would argue that the blood compartment provides a representative sample of the entire disease. This is not mentioned or discussed.

We agree with the reviewer that the current disease pathogenesis model favors a model of CLL cell recirculation between tissue and blood, but this does not imply that the blood compartment would provide a representative sample of the entire disease, especially in the context of therapy. For example, a recent study has shown that ibrutinib therapy causes a greater amount of CLL cell death in tissue compared with the blood¹. Moreover, this study revealed that a relatively small fraction of the tissue cell burden redistributes to the blood, further highlighting that the circulating cells in the peripheral blood are an inadequate surrogate of the tissue disease.

In the next sentence the authors accurately state that CLL cells proliferate in the tissues and not the blood. But whether the ctDNA testing is more suitable for picking up any proliferation-related signals than established blood or marrow tests is not clear. Proliferation goes along with transcriptional activation, but the ctDNA testing would not pick up such events. What is the relation between ctDNA and proliferation?

There is strong evidence across several studies to indicate that ctDNA is released from apoptosis of tumour cells². In the context of CLL, ibrutinib therapy has been shown to cause a greater amount of CLL cell death in the tissue compartment (where the cells are actively proliferating) compared with the blood¹. It is therefore not unexpected that ctDNA can reflect the tissue compartment where CLL cell death is most evident following treatment. Furthermore, our previously published work in solid malignancies has shown a direct relationship between increasing ctDNA release from sites of active disease progression³. In our current manuscript, numerous examples have been provided whereby ctDNA testing was more suitable for detecting sites of disease progression within the tissue compartment that were not represented by circulating cells in blood (Figure 1c and 2).

The authors continue stating that ctDNA is released directly in tissues, what is the evidence for that? I think, overall, the authors need to be more precise and less advertising about what exactly we can learn from ctDNA testing in CLL, based on the current data.

The evidence that ctDNA is released directly from cells residing in the tissues is provided by the identification of genomic changes specific to the tissue compartment (in the RS cases) which were identified through ctDNA analysis (Figure 2). In the revised manuscript we have provided additional WES of the corresponding MNL cells in these cases and show that these genomic changes could not be detected through analysis of the circulating tumour cells (Supplemental Figures 5-7).

ctDNA can be tested from PB and probably contains ctDNA from CLL cells in all disease compartments. ctDNA levels may reflect overall disease burden, but that would need to be better worked out using, for example, models to estimate overall disease burden by integrating volumetric assessment of tissue disease, the blood compartment, as well as the marrow compartment.

We agree with the reviewer that ctDNA contains DNA from CLL cells in all disease compartments; this is precisely the strength of its application in this disease. ctDNA is shed from CLL cells that are residing in both the tissues and circulation and thus provides a global representation of genomic changes across all disease sites. The referee suggests that models to estimate overall disease burden which integrate volumetric assessment of tissue disease, the blood compartment as well as the marrow compartment should be employed. As discussed above, we are unaware of a model that exists that accurately provides this volumetric assessment in model organisms leave alone human subjects. If the referee can suggest a particular model that they feel accurately provides this assessment then we would readily apply it to our data. In the absence of this, we have modified the manuscript to remove the emphasis on "overall disease burden" and describe the direct comparison between ctDNA and each test currently used to clinically assess disease burden. For instance we now specifically state that changes in ctDNA levels more accurately reflect changes in radiological disease burden than peripheral blood MNL DNA.

ctDNA further can be useful in cases where the disease mostly resides in the tissue, like in SLL, or where therapy has eliminated blood but not tissue disease. A presentation and discussion of the data in such context would make more sense to me.

We agree that ctDNA could be further useful in SLL or where therapy has eliminated blood but not tissue disease and have included further discussion to encompass these points (page 5).

To state that ctDNA testing is minimally invasive is too advertising; any other blood testing which can give us valuable information is equally minimally invasive.

It is not our intention to be advertising. We have no commercial interest or conflicts to declare. We have rephrased this in the manuscript and removed mention of the "minimally invasive" nature of ctDNA testing.

The authors answer my question of ctDNA testing versus more traditional methods of disease monitoring by stating "In contrast, total tumor burden assessment by ctDNA accurately reflected the decrease in CLL burden, and was capable of "looking past" the artificial rise in white cell count" when talking about ibrutinib therapy and transient lymphocytosis. This statement is based on declining ctDNA levels during ibrutinib treatment. Following that argument one could say that cytokines which rapidly decline after start of ibrutinib therapy also reflect disease burden, and measuring such cytokines would be more simple than ctDNA testing.

We respectively disagree with this statement. Fluctuations in cytokine levels may be affected by many different parameters such as infection or inflammation and cytokine levels are neither specific nor suggestive of changes in tumour burden. In contrast, ctDNA is specially monitoring acquired DNA mutations associated with the malignancy and changes in ctDNA levels are therefore specific to changes in that malignancy. There is a wealth of evidence to support this statement and show that ctDNA levels correlate with disease burden across many different cancer types⁴⁻⁷. As by definition, ctDNA originates from tumour cells, it is therefore not unexpected that levels of ctDNA would be linked to the disease burden.

But the main issue with this statement is that the authors have not formally looked at disease burden and do not prove that ctDNA levels reflect tissue disease burden. First, because the lymphocytosis in ibrutinib-treated patients is not artificial, it is very real and it may represent all or only a fraction of the tissue disease which is being shifted from the tissues into the blood. If all tissue CLL cells are mobilized into the blood, then declining ctDNA levels would be an inaccurate representation of overall disease burden. Second, the model therefore lacks an integrated approach of connecting blood, marrow, and tissue compartment CLL burden with the ctDNA data. Either add such model data, or tone down the statements and interpretation of the data, as suggested above.

As this is the third occasion that the referee has raised this point it is worthwhile stating once again that to the best of our knowledge there is no such model, which accurately provides a volumetric assessment of disease burden across blood, bone marrow, lymph nodes and extramedullary sites.

The standard assessment criteria, which remain the international agreed benchmark for all disease assessment in CLL is that published and recognised by the international working group on CLL (iwCLL)⁸. The referee appears to be aware of a new model of assessment that they would like us to use. If this model has been published and has been shown to be of greater value than the accepted international standard we will then gladly benchmark our ctDNA studies against this model. In the absence of this evidence we have chosen to follow the accepted international standard and have clearly stated this in the manuscript. As described above, we have also modified the manuscript to remove the emphasis on "overall disease burden" and describe the direct comparison between ctDNA and each test currently used to clinically assess disease burden.

The authors state on page 4 that ctDNA levels declined during ibrutinib therapy, unless there was an increase in disease burden radiographically. That would be highly unusual, CLL patients progress on ibrutinib at early time points only if they have Richter transformation. In how many patients was the latter seen? Were these Richter cases?

There was only one patient who exhibited a transient increase in disease burden following commencement of ibrutinib (CLL012, Supplemental Figure 4) before showing an excellent response to therapy. This transient increase in lymph node burden was assessed through clinical examination by the treating clinician, not radiologically, and this has now been explicitly stated in the revised manuscript. We agree that this clinical scenario is highly unusual, and was only evident in a single case.

Minor:

On page 3 the PI3K inhibitors should also be mentioned, given that idelalisib is FDA approved.

Idelalisib has now been included in the introduction on page 3.

Reviewer #4 (Remarks to the Author):

Sinha et al report an interesting paper describing the use of circulating tumor DNA (ctDNA) in CLL. Circulating tumor DNA is an important and emerging topic, with significant potential clinical utility. ctDNA has not been studied in this specific disease before, but has been studied in other B cell malignancies, such as non-Hodgkin lymphomas. The authors use a number of methods to describe the biology of ctDNA, including targeted NGS, digital PCR, and WES/ low depth WGS in 3 patients with Richter's syndrome.

While provocative, there are a number of points requiring further clarification, as well as limitations to the manuscript in its current form. Specifically, the authors make two major claims in their paper:

- 1) That ctDNA is more sensitive for detection of disease than MNL DNA (confirming prior results in NHL from Johns Hopkins, Stanford, and NCI that should be cited [PMIDs 21399237, 25887775, 25842160]), and
- 2) that ctDNA reflects the genomic evolution of CLL into Richter's Syndrome better than alternative methods.

Regarding claim 1):

One of the strengths is the specific focus on CLL and the number of patients and samples for this specific assessment. This does lend more credence to the claim that tumor DNA is more detectable as ctDNA than in MNLs. However, this comparison was made across many different time-points, including both pretreatment and after administration of Rituximab or Obinituzumab, which are targeted CD20 antibodies. These agents are expected to rapidly remove any

leukemia cells from circulation, and therefore, at these timepoints, it is less surprising that ctDNA is superior to MNL DNA for detection of disease. The authors should clarify which timepoints were used for the samples for comparison between ctDNA and MNL (ie: how many samples were from before therapy and how many were after therapy, and was the difference between ctDNA and MNL DNA seen at all times?)

We thank the reviewer for highlighting the strengths of the data with the increased number of cases now included in the manuscript. The papers describing the value of ctDNA in NHL have now been cited in our revised manuscript. In relation to our comparison of ctDNA versus MNL DNA in CLL, we have now separated our analysis according to those samples collected at the time of commencement of a new line of therapy (either CD20 antibody, ibrutinib or venetoclax) versus those collected following treatment (Supplemental Table 3). A clear difference in detection rates between ctDNA and MNL DNA was seen both prior and following treatment. However, this difference was most evident in the post treatment setting. These findings support the reviewer's statement that ctDNA is superior to MNL DNA for detection of disease after treatment, particularly with agents such as obinutuzumab and venetoclax which rapidly clear the circulating disease. Importantly, this highlights a key strength of ctDNA analysis for monitoring disease in CLL in the era of these potent disease-modifying agents. One of our main conclusions is that these novel therapies have dramatically changed the natural history of CLL and exposed limitations in current disease monitoring strategies. As these therapies are likely to be used more widely in the years to come it is important to have new analytical methods to follow disease activity that overcome some of the limitations in current disease monitoring strategies.

Additionally, the authors attempt to demonstrate that ctDNA better reflects radiographic disease burden than MNL DNA (figure 1d). This is an important point. However, it is not clear how the change in radiographic burden was calculated (ie: which timepoints), and how many of the points represent unique patients versus patients with multiple scans and multiple ctDNA measurements (that is, does this n=38 mean 38 individual patients?). There is a reference to the "maximal value" or radiographic disease burden in the legend – how is this determined? Is this from pretreatment CT or PET/CT, and were the quantitative measures calculated using established criteria (eg, Cheson criteria) as relevant for trial endpoints? A more detailed description of the methods is needed, and an accompanying table in the supplement of the raw data would be useful.

We thank the reviewer for these helpful suggestions. We have now included an additional table in the supplementary information (Supplemental Table 2), which includes the raw data, and highlights the serial timepoints that were used in each individual for the data analysis. A more detailed description of the radiological assessment has also been provided in the supplementary methods. All imaging was reviewed in a blinded fashion by an experienced radiologist. The imaging results were reported as the sum of the product of the perpendicular diameters (PPD) of multiple nodes as per the iwCLL guidelines

and Cheson criteria⁸. The "maximal value" in each case was defined as the maximum sum of PPD at any time point.

Separately, seeing that response criteria in CLL typically do not primarily rely on radiographic assessments, the authors should relate their findings in assessing the fidelity of ctDNA vs MNL as correlates of more traditional CLL response criteria.

Assessment of treatment response in CLL is routinely performed according to iwCLL guidelines, which include clinical examination (to assess lymphadenopathy), peripheral blood lymphocyte count and bone marrow assessment. In each individual case and across the series, we have related our ctDNA findings to treatment response assessed using these traditional measures of response. In addition we have also included the comparison of ctDNA to radiological assessment, in order to provide an additional objective measure of disease burden within the lymph node compartment.

Regarding claim 2):

The authors also claim that ctDNA can reflect clonal evolution in CLL, and provide evidence from 3 cases that support this claim, examining emerging copy number alterations and emerging SNVs. This is interesting and compelling data, however, this conclusion is based on a low number of cases (3 patients with Richter's Syndrome of 25 total patients). The generalizability of this finding is limited, given the low number of cases. Furthermore, there does not seem to be any overlying new biological finding here, other than that novel mutations can be seen in the plasma. More detailed information should be provided as to what these novel mutations are. Was any pattern of emerging mutations seen?

The authors also frame their introduction around the importance of novel agents in CLL, such as Ibrutinib and Venetoclax. In fact, all of the patients in this study received one of these drugs. However, there is no attempt to link their findings from WGS and WES in the three patients with Richter's Syndrome back to their novel therapy. No major conclusions are possible regarding the biology of CLL after treatment with Ibrutinib and Venetoclax, which is potentially due to the low number of patients (n=3) and anecdotal nature of this data. Therefore, the biological novelty of the study is limited.

We thank the reviewer for highlighting our interesting and compelling data regarding clonal evolution in CLL. We appreciate that we have only reported 3 cases but as the referee understands Richter's transformation is rare and our aim was not to perform a comprehensive analysis of the genomic events associated with Richter's transformation in CLL but rather to demonstrate the value of ctDNA in detecting this important clinical event. Moreover, as this study focuses on individuals with CLL who received either Ibrutinib or Venetoclax, two novel agents that have only just begun to be used broadly in CLL, there are very few patients available to study who have developed Richter's transformation on these novel therapies. We have one of the largest cohorts of patients in the world who have received Venetoclax for CLL⁹ and these cases represent some of the first patients who developed Richter's transformation on these agents. Whilst we

report only 3 patients in this series, the small number of cases does not limit the generalizability of the findings in demonstrating that clonal evolution in CLL can be detected through plasma ctDNA analysis. Whilst the use of ctDNA analysis to monitor clonal evolution has been demonstrated in the context of solid malignancies, our data represents the first analysis to show the potential of this technique in CLL or indeed any B-cell malignancy to study patterns of genomic evolution in patients receiving novel therapies. Our current knowledge of resistance mechanisms to ibrutinib and venetoclax are still very limited and here we show the advantage of using ctDNA analysis as a method to study resistance, which could now be used across larger prospective series of patients, treated with these agents in future clinical trials.

Ibrutinib and Venetoclax are both agents approved in CLL patients harboring 17p deletions spanning TP53, which is frequently a target of deletion during clonal evolution and clonal selection of CLL during therapy. It appears that this lesion was present nearly all of the patients profiled by the authors, ostensibly as detected by FISH on tumor-rich specimens as the basis for eligibility. The authors should comment on the noninvasive genotyping of 17p deletions using ctDNA in this same cohort, whether by targeted amplicon sequencing, WES, and/or low pass WGS.

The targeted amplicon sequencing approach that we have employed in this analysis is not optimal to assess copy number alterations. However, in the 3 cases that underwent WES and low pass WGS, copy number alterations were readily detected in plasma including the 17p deletion in the 2 cases that harbored this alteration. We have added an additional comment in the manuscript about the detection of 17p deletions noninvasively (page 5).

Furthermore, and perhaps more importantly, the authors do not show WES data from the mononuclear cells (MNL) at the time of Richter's Syndrome transformation. As the authors state that ctDNA, and not MNL DNA, accurately represents the whole body experience of disease, this seems like an important control (ie: do the emergent mutations that are seen in Richter's appear in the MNL?) The authors should include WES from the mononuclear cells at these timepoints.

We thank the reviewer for this important suggestion and have now included the WES and targeted amplicon sequencing data from the MNL cells at the time of Richter's transformation in each of the 3 cases (Supplemental Figures 5-7). This data highlights that the emergent mutations seen in the Richter's were not detectable through analysis of the MNL DNA.

Finally, the authors demonstrate two forms of clonal evolution at the time of Richter's Syndrome – linear evolution (CLL004 and CLL054) and branched evolution (CLL022). Are these types of evolution specific to Richter's Syndrome, or can they be seen in relapsing CLL without RS? There is no comparison to relapsed CLL without Richter's Syndrome. Inclusion of patients with relapsing CLL without RS in the WGS/WES analysis presented in Figure 2 would add to the biological insight of this work, if such data is available.

We agree with the reviewer that this data would be useful in providing further biological insights. Unfortunately, as the referee appreciates, relapsed CLL (without Richter's transformation) in the context of these novel therapies is rare and none of our patients have thus far relapsed without Richter's transformation.

Additional minor comments:

- Line 80-82 – “somatic driver mutations were identified in 25/32 individuals in both matched MNL and plasma DNA samples” – Table S1 does not contain the information as to which fraction the mutation was identified in, or the AF. It would be useful to know if the driver mutations were identified in MNL, plasma, or both.

We apologise for this oversight. This information has now been included in Supplementary Table 1.

- Line 129-133 – (Figure 1 C) “Case CLL001 provides an example of this scenario, where despite evidence of a significant treatment response to obinutuzumab in PB and BM, the patient developed progressive lymphadenopathy. This was paralleled by a sharp rise in ctDNA highlighting the ability of ctDNA to reflect nodal disease progression following treatment.” I am not sure that this figure supports this statement. The patient does appear to have a large spike in ctDNA from ~day 100 to ~ day 300 (ctDNA AF increases ~10x), however, the nodal disease burden (grey area plot) does not appear to increase nearly as much (increase from ~12mm² to 15mm² of nodal disease). The authors claim that the spike in ctDNA represents this change in radiological disease burden, but these seem to be on different orders of magnitude. Could there possibly be an alternative biological reason for the increase in ctDNA? This statement seems too strong as is.

The dynamic range in ctDNA is far greater than the changes that can be appreciated by radiology so we would not necessarily expect that the respective change in each parameter would be of the same order of magnitude. Nevertheless, in this case, the change in ctDNA mutant allele fraction was from day 117 (<1%) to day 292 (4%) and the corresponding change in the radiological burden was from day 175 (11mm²) to day 279 (14mm²) (i.e. 3% increase).

- Figure 2b and 2e (copy number plots from WGS of plasma at time of Richter's) – these plots are attempting to show novel copy number alterations at the time of Richter's transformation from low coverage WGS of the plasma (areas shown in red). However, the copy number plots themselves are not impressive, as the red areas do not by eye appear to be copy number altered compared to the black areas.

We have now changed the colour of the copy number plots to highlight the areas previously shown in red, which were difficult to visualise in the previous version

of the figure. All copy number calls highlighted in the plots were identified through the QDNASeq analysis algorithm (see below).

In the methods, the authors state that copy number analysis was performed using QDNASeq developed for tumor specimens. The performance of this method on ctDNA has not been established, especially in terms of detection limits and Sn/Sp. More details as to how copy number alterations were actually called, thresholds used, etc should be provided.

QDNASeq is a set of algorithms that calls copy number alterations (CNAs), after a combined correction for sequence mappability and GC content and removal of blacklist areas of problematic genome regions. It calls copy number aberrations for data using a six state/class mixture model. For our study, the genome was divided up into 15kb non-overlapping bins for analysis and the rest of the parameters were default for QDNASeq. By incorporating more than three classes, which is the basis of many other CNA algorithms, it improves detection of single copy gains and amplifications. The hierarchical mixture model allows for variability within gain or loss levels, thereby accounting for (unknown) effects, such as clonality and purity, that cause aberrations to result in non-constant log-ratio levels. The heart of the QDNASeq's CNA calling algorithm is an iterative Expectation-Maximization algorithm that estimates unknown parameters to determine CN state of a genomic segment. The algorithm used for calling a CN state within QDNASeq is derived from CGHcall¹⁰.

One of the novelties of the data presented in this manuscript is the fact that we use LC-WGS to track CNAs in plasma. We and others have previously done this with exome-sequencing, which requires a higher depth of sequencing increasing the cost and decreasing the broad application. In contrast LC-WGS of plasma provides excellent resolution of CNAs and is comparable to the data obtained via exome-sequencing (see figure below). As proof of principle we have attached below the high concordance seen between the copy number calls from the WES data on matching tumour material in case 54, provided orthogonal validation of the copy number calls made from the plasma LC-WGS data.

CLL054

Regarding the clonal evolution of CLL during Richter's Syndrome, the authors state that "In ... tumor tissue WES also revealed a high proportion of shared single nucleotide variants (SNVs) between the baseline and RS samples ... with several new SNVs emerging at the time of RS ... In each case, many of the shared and RS-specific SNVs were also identified in plasma, and several of the RS-specific SNVs were also seen in plasma at baseline (Fig. 2c and 2f)". However, the heatmaps shown in 2c and 2f do not have mutation or gene labels. These should be added to the heatmaps (likely with the heatmaps rotated 90 degrees), so that the emergent mutations can be more specifically seen.

We have now included the same heatmaps, with added data from exome analysis of the peripheral blood MNL in the supplementary information (Supplemental Figures 5-7). They have been rotated and include the gene names (due to space constraints we could not include this information within the main Fig. 2).

- There are only 25 cases in Table S1, but in the supplementary methods section the authors mention 32 patients. The 32 patients likely include patients in who targeted sequencing did not reveal mutations. Was there anything special about

these 7 patients, as opposed to the 25 where mutations were seen? The clinical information for these patients should be made available.

The details of these patients have now also been included in Supplementary Table S1.

- Figure 2 describes multiple subclones in patients with CLL and emerging clones in Richter's Syndrome. Are the sizes of the circles in figure 2a, 2d, and 2g meant to represent the sizes of each clone? If so, how were the sizes of these circles determined? Also, it appears that the heatmaps of mutations in 2c, 2f, and 2j are clustered. How was this clustering performed?

Figures 2a, 2d and 2g have been used to represent the contrasting patterns of linear versus branched evolution in these cases, but the size of the circles is not a direct representation of the size of each clone. No clustering has been performed for the heatmaps in 2c, 2f and 2j. The mutations have simply been ordered as follows; (i) all mutations present with the first sample have been represented first, (ii) followed by all mutations present within the second sample, and so forth.

References

1. Wodarz, D., *et al.* Kinetics of CLL cells in tissues and blood during therapy with the BTK inhibitor ibrutinib. *Blood* **123**, 4132-4135 (2014).
2. Snyder, M.W., Kircher, M., Hill, A.J., Daza, R.M. & Shendure, J. Cell-free DNA Comprises an In Vivo Nucleosome Footprint that Informs Its Tissues-Of-Origin. *Cell* **164**, 57-68 (2016).
3. Murtaza, M., *et al.* Multifocal clonal evolution characterized using circulating tumour DNA in a case of metastatic breast cancer. *Nat Commun* **6**, 8760 (2015).
4. Bettegowda, C., *et al.* Detection of circulating tumor DNA in early- and late-stage human malignancies. *Science Translational Medicine* **6**, 224ra224 (2014).
5. Dawson, S.J., *et al.* Analysis of circulating tumor DNA to monitor metastatic breast cancer. *N Engl J Med* **368**, 1199-1209 (2013).
6. Roschewski, M., *et al.* Circulating tumour DNA and CT monitoring in patients with untreated diffuse large B-cell lymphoma: a correlative biomarker study. *Lancet Oncol* **16**, 541-549 (2015).
7. Newman, A.M., *et al.* An ultrasensitive method for quantitating circulating tumor DNA with broad patient coverage. *Nat Med* **20**, 548-554 (2014).
8. Hallek, M., *et al.* Guidelines for the diagnosis and treatment of chronic lymphocytic leukemia: a report from the International Workshop on Chronic Lymphocytic Leukemia updating the National Cancer Institute-Working Group 1996 guidelines. *Blood* **111**, 5446-5456 (2008).
9. Roberts, A.W., *et al.* Targeting BCL2 with Venetoclax in Relapsed Chronic Lymphocytic Leukemia. *N Engl J Med* **374**, 311-322 (2016).
10. van de Wiel, M.A., *et al.* CGHcall: calling aberrations for array CGH tumor profiles. *Bioinformatics* **23**, 892-894 (2007).

Reviewers' comments:

Reviewer #3 (Remarks to the Author):

Thank-you for addressing my questions and for toning down some of the statements. You asked about models for disease burden quantification, which is detailed in the publication by Wodarz listed as reference#1 in your rebuttal. This is an attempt to quantify tissue and blood disease burden, with serial marrow testing that compartment could be added. It might be worthwhile comparing ctDNA with such a model in a future study.

Thank-you also for disagreeing on my comment about cytokines. You seem to be under the impression that these are nonspecific markers, but some cytokines are CLL-derived cytokines which are highly specific and are often used as correlative markers in CLL studies with novel agents that target BCR signaling. A more balanced response would have been better.

Reviewer #4 (Remarks to the Author):

1st comment: extended comparison analysis of ctDNA vs. MNL

- Adding Suppl. Table 3 helped clarifying the performance of ctDNA vs. MNL profiling.
- I disagree with the statement "A clear difference in detection rates between ctDNA and MNL DNA was seen both prior and following treatment". According to Suppl. Table 3, there is almost no difference at pretreatment time points (1 vs. 0 not detectable samples), while a clear difference is seen in post-treatment time points. If the authors were to do a Fisher's test, it would "clearly" (and statistically) be no different pre-treatment. They need to clearly state in the text that for pretreatment detection of disease, ctDNA and MNL DNA were statistically equivalent. This needs to be clearly addressed in the main text.

2nd comment: Clarification and more detailed description of methods used for assessment of radiographic tumor burden

- Suppl. Table 2 with raw data regarding tumor burden measurements was added.
- Methods section includes now detailed description on how radiographic assessment was performed.
- However, the authors should be more explicitly state that their n = 38 points is only from 12 patients total
- Separately, Table S2 has a significant problem, in that the authors rounded everything to two significant figures, so many allele fractions are 0.00. Did they mean that these are undetected (it seems highly unlikely that this is the case). Please either label these as undetected or give enough significant figures that readers can see the allele fractions.

3rd comment: Assessment of CLL response

- One of the concern raised in the last review, and a major concern of other reviewers, was that CLL response was assessed only by radiographic response criteria, although this cannot mirror the whole picture of tumor burden in CLL. It is understandable that the authors tried to find objective criteria to quantify response and that this can be done assessing radiographic images. Also, PBL counts are now provided in Suppl. Fig. 2. However, the authors state in their comment in this paragraph, "Assessment of treatment response in CLL is routinely performed according to iwCLL guidelines, which include clinical examination (to assess lymphadenopathy), peripheral blood lymphocyte count and bone marrow assessment. In each individual case and across the series, we have related our ctDNA findings to treatment response assessed using these traditional measures of response." There is no information about BM infiltration according to NCI or iwCLL criteria (% lymphocytes in BM, focal infiltrates). Please also provide this info to complete the picture of tumor burden.
- This reviewer would suggest as another, but certainly not perfect, objective measure for tumor burden to use levels of B2M for comparison with ctDNA and MNL AF.

4th comment: New evolving mutations in RS

- Information about new evolving mutations is now provided in Suppl. Figs. 5-7
- Since the authors ignore the question whether there are recurrent mutations evolving during Richter's transformation or a genetic pattern associated with this process, we assume that there is no such relevant novel biological finding. This should be clearly stated in the main text.

5th comment: noninvasive detection of 17p deletion

- this comment was sufficiently answered

6th comment: Adding MNL sequencing data

- this comment was sufficiently answered

7th comment: WGS/WES data on CLL relapsing patients without RS

- this comment was sufficiently answered

9th comment: The discordance between how much ctDNA goes up vs how much tumor volume goes up in CLL001, 1c:

- The author's analysis here seems very unconvincing. Firstly, 11mm^2 to 14mm^2 is not a 3% increase. That is a 27% increase ($3/11 = 0.27$). More importantly, saying ctDNA AF went from "<1% to 4%" is a "3% increase" is not accurate. The AF went from what looks like $\sim 0.5\%$ to 4%, or a 10x increase, or a 1,000% relative increase. At the same time, tumor cross-sectional area went from 11mm^2 to 14mm^2 , which in linear dimension can be something like $3\text{mm} \times 3.6\text{mm}$ to $3.7\text{mm} \times 3.8\text{mm}$ (these numbers are selected arbitrarily as an example). This is a very small change in disease burden, and in fact would not even meet the definition of progression per the Cheson Criteria / International Harmonization Project criteria.

Furthermore, the statement "the dynamic range in ctDNA is far greater than the changes that can be appreciated by radiology" is not clearly supported. Looking at table S2, the Radiology burden ranges from ~ 1.98 to 122.2mm^2 , and plasma AFs range from <1% to 63%. These seem to be on approximately same orders of magnitude of dynamic range.

Do the author's really mean mm^2 for their radiology measurements? These tumor volumes seem extremely low. Can you confirm that these are the right units, or should it be cm^2 ?

Finally –the authors should not round allele fractions to 0.00 in table S2, or anywhere else in the manuscript, unless they mean the variants were undetected. Please include enough significant figures that we can tell what the allele fraction is.

11th comment: More detailed description of QDNASeq

- The authors provide more details about CN calling with QDNASeq in the rebuttal but missed to do so in the actual methods section. It would be nice if this were explained in the main text since other research groups might want to recapitulate this work

All other minor comments have been sufficiently answered

Response to Reviewers' comments:

Reviewer #3 (Remarks to the Author):

Thank-you for addressing my questions and for toning down some of the statements. You asked about models for disease burden quantification, which is detailed in the publication by Wodarz listed as reference#1 in your rebuttal. This is an attempt to quantify tissue and blood disease burden, with serial marrow testing that compartment could be added. It might be worthwhile comparing ctDNA with such a model in a future study.

Thank-you also for disagreeing on my comment about cytokines. You seem to be under the impression that these are nonspecific markers, but some cytokines are CLL-derived cytokines which are highly specific and are often used as correlative markers in CLL studies with novel agents that target BCR signaling. A more balanced response would have been better.

We thank the reviewer for their time and expertise in reviewing the manuscript, which has now been substantially improved with their input.

Reviewer #4 (Remarks to the Author):

1st comment: extended comparison analysis of ctDNA vs. MNL

- Adding Suppl. Table 3 helped clarifying the performance of ctDNA vs. MNL profiling.

- I disagree with the statement "A clear difference in detection rates between ctDNA and MNL DNA was seen both prior and following treatment". According to Suppl. Table 3, there is almost no difference at pretreatment time points (1 vs. 0 not detectable samples), while a clear difference is seen in post-treatment time points. If the authors were to do a Fisher's test, it would "clearly" (and statistically) be no different pre-treatment. They need to clearly state in the text that for pretreatment detection of disease, ctDNA and MNL DNA were statistically equivalent. This needs to be clearly addressed in the main text.

We agree with the reviewer in relation to the pretreatment timepoints, which show concordance of ctDNA and MNL DNA rather than a difference in detection rates. We have added a statement in the main text of the manuscript (pages 4-5) to highlight this result.

2nd comment: Clarification and more detailed description of methods used for assessment of radiographic tumor burden

- Suppl. Table 2 with raw data regarding tumor burden measurements was added.

- Methods section includes now detailed description on how radiographic

assessment was performed.

- However, the authors should be more explicitly state that their n = 38 points is only from 12 patients total

As the reviewer has suggested, we have added a statement in both the methods and the figure legend of Figure 1d to explicitly state that the analysis was conducted from 12 patients (pages 13 and 19).

- Separately, Table S2 has a significant problem, in that the authors rounded everything to two significant figures, so many allele fractions are 0.00. Did they mean that these are undetected (it seems highly unlikely that this is the case). Please either label these as undetected or give enough significant figures that readers can see the allele fractions.

We thank the reviewer for recognizing this, which was an error in our revised submission. All values throughout the manuscript, figures and tables have been corrected to two decimal places. Values of 0.00 that did not meet criteria of detection as outlined in the methods section (page 10) are now denoted as ND (Not detected).

3rd comment: Assessment of CLL response

- One of the concern raised in the last review, and a major concern of other reviewers, was that CLL response was assessed only by radiographic response criteria, although this cannot mirror the whole picture of tumor burden in CLL. It is understandable that the authors tried to find objective criteria to quantify response and that this can be done assessing radiographic images. Also, PBL counts are now provided in Suppl. Fig. 2. However, the authors state in their comment in this paragraph, "Assessment of treatment response in CLL is routinely performed according to iwCLL guidelines, which include clinical examination (to assess lymphadenopathy), peripheral blood lymphocyte count and bone marrow assessment. In each individual case and across the series, we have related our ctDNA findings to treatment response assessed using these traditional measures of response." There is no information about BM infiltration according to NCI or iwCLL criteria (% lymphocytes in BM, focal infiltrates). Please also provide this info to complete the picture of tumor burden.

- This reviewer would suggest as another, but certainly not perfect, objective measure for tumor burden to use levels of B2M for comparison with ctDNA and MNL AF.

We thank the reviewer for these helpful suggestions. We have now included additional information in table S2, which details BM infiltration (including %lymphocytes in BM) and B2M levels where available. As the iwCLL criteria only requires marrow examination when the patient is suspected to be in complete remission or if there is suspected disease progression, marrow examinations

were performed infrequently in this cohort. Moreover, as B2M monitoring is not currently considered standard in CLL, this was not performed routinely in our cohort. For this reason, we were limited in the number of timepoints where this information was available and statistical comparison between changes in %lymphocyte infiltration or B2M levels and ctDNA could not be performed. However, we do recognize the prognostic value of B2M and its potential reflection of tumour burden and we agree that this warrants further investigation in future studies.

4th comment: New evolving mutations in RS

- Information about new evolving mutations is now provided in Suppl. Figs. 5-7
- Since the authors ignore the question whether there are recurrent mutations evolving during Richter's transformation or a genetic pattern associated with this process, we assume that there is no such relevant novel biological finding. This should be clearly stated in the main text.

There are no recurrent mutations or genetic patterns that are consistently observed in our cases of Richter's transformation. Therefore, as suggested by the reviewer, we have now added an additional statement in the text (page 7) to clarify this point.

5th comment: noninvasive detection of 17p deletion

- this comment was sufficiently answered

6th comment: Adding MNL sequencing data

- this comment was sufficiently answered

7th comment: WGS/WES data on CLL relapsing patients without RS

- this comment was sufficiently answered

9th comment: The discordance between how much ctDNA goes up vs how much tumor volume goes up in CLL001, 1c:

- The author's analysis here seems very unconvincing. Firstly, 11mm^2 to 14mm^2 is not a 3% increase. That is a 27% increase ($3/11 = 0.27$). More importantly, saying ctDNA AF went from " $<1\%$ to 4% " is a "3% increase" is not accurate. The AF went from what looks like $\sim 0.5\%$ to 4% , or a 10x increase, or a 1,000% relative increase. At the same time, tumor cross-sectional area went from 11mm^2 to 14mm^2 , which in linear dimension can be something like $3\text{mm} \times 3.6\text{mm}$ to $3.7\text{mm} \times 3.8\text{mm}$ (these numbers are selected arbitrarily as an example). This is a very small change in disease burden, and in fact would not even meet the definition of progression per the Cheson Criteria / International Harmonization Project criteria.

We agree with the reviewer that the magnitude of the change in the radiological

disease burden was small and did not meet criteria for progression as per the iWCLL guidelines. Clinically there was evidence of a new palpable supraclavicular lymph node measuring 1.5cm in size on physical examination (we have now included this information in the revised text, page 5). The combination of physical examination and radiological findings led to the change in treatment and the initiation of ibrutinib, which resulted in improved disease control. We appreciate that the magnitude of change between ctDNA and radiology will vary from case to case as evidenced here. Here ctDNA is measured by dPCR, which provides an absolute quantification of the number of tumour derived DNA molecules. This resolution is beyond the capacity of radiological imaging. The important conclusion we have emphasized is that whilst the magnitude of change may be different, the changes are concordant.

Furthermore, the statement “the dynamic range in ctDNA is far greater than the changes that can be appreciated by radiology” is not clearly supported. Looking at table S2, the Radiology burden ranges from ~1.98 to 122.2mm², and plasma AFs range from <1% to 63%. These seem to be on approximately same orders of magnitude of dynamic range.

We accept the reviewer's comments and as discussed above the magnitude of change in ctDNA levels and radiological burden do vary within individual cases. One of the main conclusions we have focused on in the manuscript, is the fact that the change in radiological disease burden correlates with the change in ctDNA mutant allele fraction, as highlighted in Figure 1d.

Do the author's really mean mm² for their radiology measurements? These tumor volumes seem extremely low. Can you confirm that these are the right units, or should it be cm²?

We thank the reviewer for recognizing this typographical error - the units are cm² rather than mm² and this has been corrected.

Finally –the authors should not round allele fractions to 0.00 in table S2, or anywhere else in the manuscript, unless they mean the variants were undetected. Please include enough significant figures that we can tell what the allele fraction is.

As highlighted above, we thank the reviewer for identifying this. We have denoted all values that were considered not detectable by our methodology as ND in the manuscript instead of 0.00.

11th comment: More detailed description of QDNASeq

- The authors provide more details about CN calling with QDNASeq in the rebuttal but missed to do so in the actual methods section. It would be nice if this

were explained in the main text since other research groups might want to recapitulate this work

We have expanded the description of the QDNaseq methodology in the main text as suggested (page 12).

All other minor comments have been sufficiently answered

We are very grateful to the reviewer for their input on the manuscript, which has been substantially improved through the review process.

REVIEWERS' COMMENTS:

Reviewer #4 (Remarks to the Author):

The authors seem to have addressed all my major and most of my minor concerns, and I believe that the manuscript may have improved as a result of this review.

NCOMMS-16-19361B

RESPONSE TO REVIEWERS' COMMENTS

Reviewer #4 (Remarks to the Author):

The authors seem to have addressed all my major and most of my minor concerns, and I believe that the manuscript may have improved as a result of this review.

We thank the reviewer for the time they have put into reviewing the manuscript and agree that the manuscript has improved through this process.